# Accidents, Injuries, and Safety among Artisanal and Small-Scale Gold Miners in Zimbabwe

**DOI:** 10.3390/ijerph19148663

**Published:** 2022-07-16

**Authors:** Josephine Singo, John Bosco Isunju, Dingani Moyo, Stephan Bose-O’Reilly, Nadine Steckling-Muschack, Antony Mamuse

**Affiliations:** 1Centre for International Health, University Hospital, LMU Munich, Leopoldstrasse 5, D-80802 Munich, Germany; 2Devsol Consulting, Clock Tower, Kampala P.O. Box 73201, Uganda; 3Exceed Institute of Safety Management and Technology, Kampala P.O. Box 72212, Uganda; 4Disease Control and Environmental Health Department, Makerere University School of Public Health, Kampala P.O. Box 7072, Uganda; isunju@musph.ac.ug; 5School of Public Health, University of the Witwatersrand, Private Bag 3, WITS, Johannesburg 2050, South Africa; moyod@iwayafrica.co.zw; 6Faculty of Medicine, National University of Science and Technology, Ascot, Bulawayo P.O. Box AC 939, Zimbabwe; 7Faculty of Medicine, Midlands State University, Private Bag 9055, 263, Senga Road, Gweru P.O. Box 9055, Zimbabwe; 8Institute and Clinic for Occupational, Social, and Environmental Medicine, University Hospital, LMU Munich, Ziemssenstr. 5, D-80336 Munich, Germany; stephan.boeseoreilly@med.uni-muenchen.de (S.B.-O.); nadine.steckling-muschack@klinikum-os.de (N.S.-M.); 9Department of Public Health, Health Services Research and Health Technology Assessment, UMIT-University for Health Sciences, Medical Informatics and Technology, Eduard-Wallnoefer-Zentrum 1, A-6060 Hall in Tirol, Austria; 10Klinikum Osnabrueck GmbH, Am Finkenhuegel 1, D-49076 Osnabrueck, Germany; 11Department of Geosciences, Midlands State University, Private Bag 9055, 263 Senga Road, Gweru P.O. Box 9055, Zimbabwe; mamusea@staff.msu.ac.zw

**Keywords:** accidents, injuries, risk factors, control measures, safety, artisanal and small-scale gold miners, Zimbabwe

## Abstract

Artisanal and small-scale gold mining (ASGM) employs 14–19 million people globally. There is limited research on accidents, injuries, and safety in Zimbabwe’s ASGM. This study investigates the prevalence of accidents and injuries, as well as the associated risks and existing safety practices. A cross-sectional survey was conducted among artisanal and small-scale gold miners. Data from 401 participants were analyzed using descriptive statistics and regression analysis. The prevalence of accidents and injuries was 35.0% and 25.7%. Accidents associated with experiencing injuries included mine collapses and underground trappings. The major injury risk factors were digging, blasting, being male, being 18–35 years old, crushing, and the underground transportation of workers and materials. Injuries were reported highest among the miners working 16 to 24 h per day. Participants had heard about personal protective equipment (PPE). There was training and routine inspections mainly on PPE use. Mine owners and supervisors were reported as responsible for OSH, which was mainly PPE use. Practices including the use of wire winch ropes and escape routes were rare. There was ignorance on underground mine shaft support. The mining regulations that had the potential to introduce comprehensive safety controls were not adaptable. We recommend applicable health and safety regulations for Zimbabwe’s ASGM.

## 1. Introduction

Recent global estimates assert that 14–19 million people are employed in artisanal and small-scale gold mining (ASGM) [1]. In Zimbabwe, ASGM is one of the informal sectors with a key economic role as it serves as a fundamental livelihood source [2]. This role has recently expanded as a result of persistent droughts and increasing unemployment rates worsened by the negative impacts of COVID-19.

Zimbabwe’s unemployment rate in 2019 was estimated at 16% with 76% employed in the informal sector [3]. With more than 500,000 citizens working in artisanal and small-scale mining (ASM) [4] (mainly ASGM), ASGM is a significant source of informal employment in the country. ASGM varies in its formality and legality, operating through basic, manual, or primitive methods, and is usually associated with little or no attention to health and safety [5,6]. In 2019 alone, Zimbabwe had an estimated 4124 occupational injuries with a national occupational injury rate of 4.8 per 1000 and 6.7 per 1000 for large-scale mining (LSM) [3], which was based on claims made to the National Social Security Authority (NSSA). ASGM employs more people than large-scale mines, and the prevalence of global ASM occupational injuries is approximately seven times higher than in large mines [6].

ASGM is characterized by a high prevalence of injuries and fatalities due to numerous hazards and a lack of safety standards [6]. According to McFarlane’s online media data set on ASM fatalities, 592 fatalities were reported globally in 2020, while 42 fatalities were recorded in Zimbabwe [7]. A study on occupational accidents in artisanal mining in the Democratic Republic of Congo (DRC) found the occurrence of accidents at 32.9% for the miners who were handling heavy loads [8]. Estimated injury rates in Ghana were 45.5 and 38.5 per 100 person-years in 2011 and 2013, respectively in a cohort study [9]. A study on injuries among ASGM miners in Ghana revealed experiences of catastrophic injuries due to collapsing underground mining pits [10]. Accidents in ASGM lead to injuries and fatalities of different intensities [10], including fractures, cuts, bruises, and loss of life [10,11,12,13,14]. Alcohol consumption, drug use, and smoking cigarettes at work are common in ASGM within Zimbabwe [11,12,15]. In Kenya, injuries in ASGM were associated with increased occupational alcohol and drug use [16].

Underground ASGM is widespread in Zimbabwe. Unfortunately, so are the risks associated with mine collapses and accidents [7,17]. Literature has revealed that underground mining is associated with processes requiring effective control measures [18,19]. Furthermore, underground artisanal and small-scale gold mining has been described as inherently hazardous [19,20]. Previous studies on ASGM in Zimbabwe have found compromised personal protective equipment (PPE) use, unsafe underground shafts, and low compliance with mining regulations [11,12,15]. Furthermore, a case-control study on severe occupational injuries at a mining company in Zimbabwe identified working underground, insufficient PPE use, and working shifts longer than eight hours as risk factors associated with severe injuries [14]. In Ghana, poorly supported underground shafts and poor pit designs resulted in ground failures, leading to fatalities and injuries of varying severity [10]. Studies have depicted digging, shaft sinking, underground mining, blasting, crushing, and long working hours as high-risk factors associated with accidents and injuries in Ghana’s ASM sector [19,20]. Experts have therefore acknowledged the need for support systems for underground excavations, hence the relevance of technical backing in ASGM in Zimbabwe [18].

Accidents and injuries can result in multiple losses [21], e.g., lost income, time lost to injuries, and loss of production [22,23]. Contrarily, a healthy workplace is correlated to improved safety and increased mine production [22,24]. Risk management is therefore imperative in mining safety and health [22], and integral to a substantial decline in injuries in mining [22,25]. An effort is being made by the Ministry of Mines and Mining Development, non-governmental organisations (e.g., Pact), ASM associations, and mine owners to raise awareness and improve PPE use in Zimbabwe’s ASGM sector. However, PPE use is the least effective control measure [26]. A single layer of PPE use alone is inadequate to mitigate mining risks [18,19,22,25] as illustrated in A and B of Figure 1: modified after Figure 1 in our previous article [12].

The Swiss cheese model, which simplifies the correlation between exposure to hazards, vulnerable control measures, and the likelihood of the occurrence of accidents, assumes various layers of control measures characterized by “holes” of different shapes and sizes (Figure 1), representing multiple weaknesses in the control measures, resulting in accident opportunities [21]. The ranking of the hierarchy of controls from the least to the most effective is PPE, administrative controls, engineering controls, substitution, and elimination [26]. Elimination and substitution controls are challenging for existing operations [26]. Simultaneously, engineering controls are less dependent on human effort and are effective [26]. Fogler has applied the Swiss cheese model in engineering and has labeled the successive defense layers as engineering controls, administrative controls, behavioral controls, and mitigation barriers [27]. The Swiss cheese model has successfully been applied in the formal sector including large-scale mining (LSM) [28]. Successive layers of control measures could reduce accidents and injuries in ASGM [22,28]. However, ASGM is a poverty-driven, informal, and undercapitalised sector with a low degree of professionalism [5]. Research has indicated the lack of comprehensive occupational health and safety management in ASGM which is correlated to occurrences of accidents and injuries [19,29,30,31]. Smith, therefore, argues that ASM (which includes ASGM) should be prioritized as a high-risk sector requiring public health interventions including initiatives on occupational health and safety [30].

ASM in Zimbabwe is commonly unregulated, lacking technical and management skills [5], including safety and health management [32]. Effective control measures, such as standard mine shafts, are therefore missing and PPE is the common and compromised control measure [11,12,15]. There is high exposure to hazards for workers in ASGM [6,7], as shown in A and B of Figure 1. Reasons for reduced PPE use among workers in Zimbabwe’s ASGM include unaffordability, non-compliance, and negative perceptions [11,12,15]. Missing and compromised control measures could result in multiple opportunities for accidents [21], as represented by the arrows in A and B in Figure 1. A double layer of control measures, C of Figure 1, could provide the opportunity to reduce exposure to hazards, e.g., through accident investigation that involves the management of identified causes of accidents [33,34]. Hazard identification and risk assessment is the initial step of risk control [31,32]. At the same time, training and supervision are fundamental elements of risk management [35]. Concurrently, administrative defenses (C of Figure 1) are rare in ASGM in Zimbabwe. Mining is defined as a high-risk sector in Zimbabwe [36], the mining regulations [36] therefore stipulate the required safety standards that incorporate successive defense layers [21] and an opportunity for comprehensive protection, as illustrated in D of Figure 1. However, mining regulatory standards are similar for both LSM and ASGM and there is limited capacity to comply with mining safety regulations in ASGM in Zimbabwe [37]. Scholars have found that mining regulations are ambiguous and deficient to address the dynamics of the ASGM sector in Zimbabwe [37,38]. Hence, the need for compatible mining regulations to address the needs of ASGM [37,38,39]. Although accidents and injuries are common in ASGM in Zimbabwe, there is limited research on injuries and accidents in ASGM in Zimbabwe. The aim of this study was to investigate the safety of miners involved in ASGM in Zimbabwe. Findings could be used to guide policies and relevant interventions to support the sector. The specific objectives of this study were to (1) assess the prevalence of accidents and injuries in ASGM, and (2) explore associated risk factors and safety management opportunities for ASGM in Zimbabwe.

## 2. Materials and Methods

The materials and methods in this section were published in our first article based on the same survey [12]. A cross-sectional survey was conducted through an administered questionnaire, (Appendix A) in the districts of Shurugwi and Kadoma from November to December 2020. Participants were selected by multi-stage sampling [12]. Mining areas that were actively involved in rudimentary and more mechanized mining methods were selected purposively in Kadoma and Shurugwi [12]. This was followed by a simple stratified random sampling by reshuffling names of identified sites that were involved in rudimentary and more mechanized mining methods [12]. Participants in the selected sites were chosen randomly while making sure to maintain proportional gender inclusion [12]. The target population was miners working in ASGM in Kadoma and Shurugwi [12]. Consenting adults, who were at least eighteen years old and with at least six months experience in ASGM, were included in the survey [12]. Miners who were intoxicated, who had low levels of literacy, were not comfortable filling the questionnaires with support, or were not interested in the study were excluded [12]. The purpose of the study was explained to the participants. Questions on the research were addressed and consent forms were signed before any participants completed the questionnaire.

### 2.1. Study Area

Shurugwi and Kadoma districts are in the Midlands and Mashonaland West provinces of Zimbabwe, respectively. The national survey on ASGM in Zimbabwe found that Midlands and Mashonaland West were the most active provinces in terms of ASGM mining activity density, the density of processing sites, and the number of formally registered gold milling sites.

ASGM activities in Kadoma and Shurugwi involve unlicensed groups and individuals, licensed individuals and groups, and licensed small-scale mining companies [40]. Rudimentary and more mechanized mining operations are common in Kadoma and Shurugwi. Sites practicing these types of mining were visited in Patchway, Battlefields, Sanyati, Mayflower, Brompton, and Mudzengi in Kadoma, and Wonderer and Chachacha in Shurugwi.

### 2.2. The Administered Questionnaire

Questionnaires were available in Shona and Ndebele, Appendix A. Shona, the first language of the participants, was used in the administration of the questionnaires by trained and experienced data collectors. Data collectors were trained through a participatory approach as indicated in our separate article, which is based on the same survey [12]. Questions on accidents, injuries, and underground mining were included in the administered questionnaires. Questionnaires were designed based on previous ASGM surveys in Kenya and Zimbabwe [11,41] and were available in Shona and Ndebele. The questionnaires were pre-tested among eight miners from Kadoma and Gwanda in Zimbabwe, and then modified and translated into Ndebele and Shona by experienced translators [12]. The questionnaire was approved along with the study protocol by the Medical Research Council of Zimbabwe [12]. After the participants received the questionnaires, data collectors went through the questionnaires with participants as a group, then participants independently filled out the questionnaires. More support was given as required for participants who had lower literacy abilities and were willing to participate in the study. Participants who were not comfortable participating in the study because of low literacy were not included in the study.

### 2.3. Data Analysis

Data from the questionnaires were entered, cleaned, and analyzed in SPSS version 20. Categorical data from the questionnaires were summarized using frequencies and percentages as indicated in our separate article based on the same survey [12]. The odds of experiencing injuries and accidents were assessed through cross-tabulation against relevant risks and were presented as unadjusted odds ratios (ORs). The association between experiencing injuries and predictor variables such as age, gender, and workplace roles was assessed using binary logistic regression and was presented as adjusted odds ratios (AORs). The level of significance was set at *p* ≤ 0.05.

### 2.4. Ethical Approval

The study was approved by the University of Munich Ethics Committee (Project 20-068) and the Medical Research Council of Zimbabwe (MRCZ/A/2603) [12]. In addition, consent was sought from local authorities, mine owners, and all participants [12]. Those chosen for the study participated voluntarily and signed the informed consent form prior to data collection. Questionnaires were numbered without names to ensure confidentiality [12]. The data set is available on Mendeley and is accessible upon request.

## 3. Results

The questionnaires had 401 respondents, with a response rate of 88% as indicated in our separate article based on the same survey [12].

### 3.1. Socio-Demographic Characteristics

The questionnaires had 401 respondents, for further details on socio-demographic characteristics see Table 1 in our separate article based on the same survey [12]. Seventeen percent of the respondents were women. The proportion of married miners was 51.1%, with more than 50% of the participants between 18 and 36 years old. Education background varied, with 10% receiving tertiary education, 15% had completed primary school, and 7% having no formal schooling. Knowledge and competence in ASGM were low for more than 25% of the survey participants. The migration from site to site was 28%.

The significant roles, mainly taken up by men, included digging, moving ore manually, blasting, and loading ore. Nearly 65% (*n* = 136) of the miners in Kadoma were involved in digging. More than 30% of the miners worked for more than eight hours per day. Two hundred and one of the participants worked underground. Twenty-seven percent of the participants had up to six months of experience in ASGM and had joined the sector during the COVID-19 pandemic.

### 3.2. Reported Occupational Safety and Health (OSH)

Of the 401 participants, 198 (49.4%) of the participants reported that they had heard about OSH. Thirty-six percent (*n* = 138) indicated that they had been trained on (OSH), which was mainly PPE use. Of the 14 participants who indicated their trainers, 11 (78.6) were trained by non-governmental organizations (NGOs) and 3 (21.4%) were trained by community members. However, the prevalence of injuries among the participants who had trained and those not trained was 39% (*n* = 39) and 34% (*n* = 84), respectively (OR 1.2 (0.8–2.0) *p* = 0.3). Training seemingly had no significant impact on the prevalence of accidents and injuries, which could be attributable to the ineffectiveness of training or response bias. The person responsible for safety was reported as mine owner 40.7% (*n* = 160), supervisor 21.9% (*n* = 86), gang leader 12.2% (*n* = 48), employee 21.6% (*n* = 85), and sponsor 3.6% (*n* = 14). There was a form of management and an opportunity for safety management as indicated by the roles of responsibility on safety. However, the common understanding of safety and health was the provision of PPE. PPE was found low and compromised during the same survey [12].

#### Observed Safety Practices (Observed by Chance during the Survey)

During the survey, one site in Sanyati introduced the wire winch rope. Three sites visited in Kadoma had escape routes. One site in Kadoma had a plan for a waterway to avoid shaft flooding. The mine operation, which had invested in shaft support, had a series of shafts in close proximity to each other, posing the high risk of a major shaft collapse. In addition, one site that was operated by a mining company had gathered timbering wood to transfer mining operations to a new shaft within the same site because of reported high levels of fumes and gasses that had accumulated from blasting. However, there was no knowledge on debarking the timbering wood to reduce the accumulation of gasses. At the same time, the debarking of timbering wood was observed on two sites. Focus group discussions conducted during the same survey indicated that it was expensive for the average ASG miners to invest in mine support, escape routes, and the wire winch rope [12]. Hazard identification and risk assessment were not practiced. The Ministry of Mines and Mining Development was conducting routine inspections mainly on PPE use and there were penalties for non-PPE use, an opportunity for implementation of safety and health management.

### 3.3. Reported Accidents

Accidents were reported by 140 (35%) miners. Table 2 shows the types of reported accidents and reported occurrences of accidents.

Slips, trips, and falls (STFs), hit by tools or machines, and hit by pieces of stones were the most reported accidents. The mine collapses, underground trappings, and instant deaths were also reported. The study was conducted in the rain season. The mine collapses and STFs could have been associated with the rainy season. During the same survey, breaking of the winch rope was associated with a loss of ability to work; one site had therefore introduced the wire winch rope. However, focus group discussions revealed that the wire winch rope was expensive for an average ASG miner. The types of accidents that were mostly associated with occurrences of injuries were underground trappings, breaking of the winch ropes, and shaft collapses.

Participants indicated that accident reporting could be done to the local chief (*n* = 36, 41.3%), the Ministry of Mines and Mining Development (*n* = 112, 31.5%), the police (*n* = 18, 5.1%), the hospital (*n* = 67, 18.8%), or not reported (*n* = 12, 3.4%). Actions to be taken after an accident included investigation into the cause (*n* = 206, 66%), temporary closure (*n* = 70, 22.4%) of the operation, and no action at all (*n* = 36, 11.5%). The sites visited had no registers of near-miss incidents and accidents. Accident investigations were not conducted internally by the miners. External investigations were conducted by mine inspectors (the Ministry of Mines and Mining Development) in the event of reported accidents that had associated penalties such as mine closures. Internal accident reporting and incident investigation were therefore not common.

### 3.4. Reported Injuries

One hundred and three miners indicated that they had been injured at work. The prevalence of injuries in age groups varied from 31.6% (*n* = 61) for those 18–35, to 26.4% (*n* = 33) for 36–50-year-old miners, and 16.1% (*n* = 5) for those over 50 years old (Table 4). Ever-experienced injuries were reported by 71 (38.4%) miners working underground (OR = 3.1 (1.8–5) *p* < 0.0001), which complements findings from the same survey [12]. Rudimentary and more mechanized mining categories reported combined accidents and injuries at 29.2% (*n* = 21), and 48% (*n* = 143), respectively (OR = 2.2 [1.3–4] *p* = 0.004. The type of injuries reported during the same survey were fractures (*n* = 34, 52.2%), cuts (*n* = 24, 41.4%), and bruises (*n* = 22, 37.9%), which results in injuries on hand(s) (*n* = 38, 28.1%), leg(s) (*n* = 32, 23.7%), finger(s) (*n* = 24, 17.8%), head (*n* = 19, 14.1%), and chest (*n* = 15, 11.1%) [12].

### 3.5. Risk Factors

Of the miners who had experienced workplace violence, 39.4% (*n* = 37) had ever experienced injuries at work (OR = 2.1 (1.3–3.4) *p* = 005), which complements findings from the same survey [12]. The prevalence of injuries increased with increasing working hours (Figure 2).

Injuries were reported highest among the miners working 16 to 24 h daily. Among the 44 participants who were working for >16–24 h daily, 40.9% (18) had been injured before. Of the miners who were involved in more mechanized and underground mining, 95.7% (*n* = 44) were working for 16 to 24 h daily. The prevalence of injuries among the artisanal small-scale gold (ASG) miners who were working underground was 57.7% (*n* = 45) and 65.9% (*n* = 29) for >8–16 and >16–24 daily working hours, respectively. The distribution of injuries according to experience in ASGM is shown in Figure 3.

The injuries decreased to 22.1% (*n* = 31) from 24.7% (*n* = 23) after 6–12 months of experience and were more than 30% after more than five years of experience in ASGM, which could be attributable to shifting to more hazardous roles such as blasting with increased experience and lack of training. Due to COVID-19, there were an increased number of new workers without proper training. The regulation on the limited number of workers per site during COVID-19 restrictions could have forced more experienced miners to take on new unfamiliar roles. Training was not associated with decreased prevalence of injuries as presented above. There was no association between experiencing injuries and the level of education or marital status, which could be attributed to response bias. The association between workplace roles and injuries is shown in Table 3.

The unadjusted odds of experiencing injuries were at least double for the miners involved in moving ore manually, loading, digging, and blasting. Milling was associated with accidents and loss of fingers during the same survey [12].

### 3.6. Association between Ever-Had Injuries at Work and Exposure to Risk Factors

Of the 401 miners, 370 participants who responded to the question were included in binary regression analysis; 31 participants did not respond to the question, and 103 (25.7%) had experienced injuries (Table 4). The regression model explained 27% of the variability between exposure to risks and experiencing accidents and injuries (R^2^ 26.5%, *p* < 0.0001). The association between ever-experienced injuries at work when exposed to risk factors, is shown in Table 4 below.

After adjusting for other variables in the model, the odds of ever-had injuries were more than nine times higher for crushing and blasting. Blasting was associated with mine collapses and fatalities (Table 2). The odds of experiencing injuries were more than four times more for men (AOR = 4.3 (1.4–13.6)) than for women. The likelihood of experiencing injuries was twenty percent more for the 18–35-year age group (AOR = 0.2 (0.07–0.9)) compared to the >50 age group. Simultaneously, the odds of experiencing injuries were five times more during the transportation of miners and ores to and from underground shafts (AOR = 4.9 (2.1–11.2), AOR = 0.04 (0.005–0.3)). Focus group discussions during the same study revealed a tendency to use a worn-out winch rope, which was evidenced by a case of a miner who got injured and lost his ability to work when a worn-out winch was used when he was being transported from the shaft [12]. Miners reported the use of other intoxicating substances, as shown in Figure 4 below.

Among the 97 participants who reported taking intoxicating substances, 89 admitted to alcohol consumption, five used drugs, and marijuana, while three reported a combination of drugs and alcohol use. Alcohol use and smoking were observed on mining sites. One respondent who reported taking marijuana described marijuana as ‘mupapfungwa’, i.e., wisdom source, a perception that could strengthen drug use in ASGM. No further analysis was conducted on the direction of association between alcohol drugs and injuries, the question on alcohol consumption and drug use was secondary to the question on smoking and there was a low response. Further analysis is recommended for further pieces of research.

## 4. Discussion

This cross-sectional survey found the prevalence of accidents and injuries in ASGM in Zimbabwe at 35.0%, and 25.7%, respectively. Accidents that had high risks of experiencing injuries were slips, trips, and falls (STFs), flying particles, mine shaft collapses, and underground trappings. STFs and mine collapses were common in the rainy season. The majority of the participants had heard of OSH, which was mainly PPE use. Training and routine inspection on PPE use was found. Mine owners, supervisors, employees, and sponsors were reported as responsible for OSH, which was mainly PPE use, an opportunity for risk management. The associated risks included long working hours, alcohol and drug abuse, and underground mining. The other factors associated with a high risk of experiencing injuries were being male, being in the age group 18–35 years, digging, blasting, loading, the transportation of miners and materials from shafts, and crushing. Figure 1 illustrates different settings of high exposure to hazards, reduced exposure to hazards, and comprehensive protection. ASGM in Zimbabwe is characterized by high exposure to hazards [11,12,15]. This section discusses the prevalence of accidents and injuries, the associated risk factors, and the opportunities to improve safety in ASGM to achieve reduced exposure to hazards and comprehensive protection.

During the cross-sectional survey, 35.0% of the participants reported ever-experienced accidents at work, which is comparable to similar studies. For example, 32.9% of respondents in the DRC’s ASM reported accidents when handling heavy loads [8]. The odds of experiencing accidents and injuries were more than double for more mechanized mining operations compared to rudimentary operations, which could be attributed to long working hours and underground mining, without successive layers of control measures as further discussed below. Alcohol consumption and drug use at work were common with no control measures, this concurs with previous literature [11,12,15]. Working under the influence of alcohol and drugs is a significant determinant of accidents and injuries in ASM [8,16]. In Kenya, the prevalence of accidents in ASGM was found to be higher among high-risk drug users, 34.2% (*n* = 25) compared to non-drug users 13.6% (*n* = 11) (*p* = 0.001) [16]. There was a perception that drug use at work instilled wisdom among ASG miners, a harmful myth that could strengthen risky behavior in Zimbabwe’s ASGM [32]. Hence, the need to reduce exposure of ASGM to hazards through relevant public health interventions such as peer counseling and awareness raising to mitigate the risk of workplace alcohol consumption and drug use.

The prevalence of ever-experienced injuries, i.e., 25.7%, surpassed Zimbabwe’s 2019 LSM rate of 6.7 per 1000, which was based on claims from LSM corporations that were submitted to NSSA [3]. LSM workers also receive health and social security coverage, while such services are not accessible for workers in Zimbabwe’s ASGM [11,12]. Outside of Zimbabwe, ASGM injuries are also much higher than those working in LSM [6], as seen in Ghana, where approximated injury rates were 45.5 per 100 persons in 2011 and 38.5 per 100 person-years in 2013 [9]. The odds of experiencing injuries were higher for men than women while the prevalence of injuries decreased with age, confirming previous findings [10]. Since the 18–35 age group was at a higher risk of experiencing injuries, the higher proportion of ASG miners aged 18–35 during the survey implies an increased prevalence of injuries in Zimbabwe’s ASGM during the study period.

Underground artisanal mining is commonplace in Zimbabwe. This type of mining requires more technical processes, such as mine support, drilling, blasting, and loading [18], and is associated with increased risks compared to surface mining [18,19]. Injury risk factors associated with underground mining included long working hours, more mechanized mining processes, and high-risk roles, which resulted in increased odds of experiencing injuries, echoing existing literature [8,12,18,19]. In Kenya, the miners’ odds of experiencing injuries were 2.6 (*p* = 0.002) times higher for those working more than eight hours per day compared to the miners working for less than eight hours per day [16]. Workers with high-risk underground mining roles, i.e., blasting, crushing, loading, and underground transportation of people and materials to and from underground shafts had increased odds of experiencing injuries (as documented in previous literature) [18,31]. In Ghana, mine pit collapse was the most frequent cause of accidents associated with injuries, followed by blasting injuries [10]. Ground failures, due to unsupported or poorly supported shafts and poor pit design, also led to fatalities and injuries of varying degrees in Ghana ASM operations [19]. The ground failures were attributed to a lack of planning, the unfamiliarity with rock strength and stability, and an incorrect choice of mining methods stemming from a lack of technical knowledge and experience [19]. Mine support is therefore defined as one of the top priorities in mining safety [18]. However, artisanal, and small-scale underground mining is usually associated with sub-standard mine support [11,12,15,19]. While the mining regulations have set the standards for underground mine support, substandard and unsafe mining pits are common in Zimbabwe’s ASGM [11,12,15]. As presented above, there is a need for adaptable ASM regulations, as well as technical support, as further presented.

Since underground ASGM mining is a high-risk sector and ASGM in Zimbabwe is associated with injuries and fatalities [7,10,17,18,19,20,32], there is a need for accident prevention control measures to reduce exposure of ASG miners to associated risks. Internal hazard identification and risk assessment and incident reporting and investigation were not found in this study. Hazard identification and risk assessment is the initial step toward risk control, which guides the development of prevention strategies [35]. Furthermore, for every fatal accident, there are 10 serious accidents, 30 minor accidents, and 600 near-misses, 1:10:30:600 [42]. Apparently, accident investigations were conducted by mine inspectors and were associated with penalties. Accident reporting was therefore not common. The behavior of concentrating on one fatality while neglecting 600 near-misses, which provide the opportunity to prevent the fatality, is harmful [34]. Hence the importance of an internal incident investigation that focuses on near-misses (six hundred near misses to prevent the fatality) for accident prevention in ASGM in Zimbabwe. The motivation of the sector could facilitate the uptake of such safe practices, thereby reducing exposure to hazards. The business and market-centered approach has been used to improve returns and incentivise and motivate protective safe practices in ASGM [43,44].

In addition to compromised PPE use, training is fundamental to reducing weaknesses in individual control measures and establishing successive defense layers of protection. A low competence in mining was found as demonstrated by a lack of knowledge on mine support. In addition, longer work experience was not associated with a lower prevalence of injuries, which confirms previous findings [10] that attribute a lack of training and exposure to riskier roles, such as blasting, with increased ASGM experience. Low levels of literacy and limited knowledge on mining are common among ASGM miners [5,7]. However, in Ethiopia, increasing work experience was associated with decreasing non-fatal injuries [13]. This research was conducted during the COVID-19 pandemic and so this was a time that could have witnessed more experienced workers forced to take on new and unfamiliar roles due to restrictions on the number of workers allowed per site. Simultaneously, a lack of training is also common in artisanal and small-scale mining (ASM), as is confirmed in DRC where a lack of training was also found to be one of the determinants of accidents in ASM [8]. Although some participants indicated that they had trained on PPE, there was no association between training and a reduced prevalence of accidents. The training was mainly conducted by NGOs. Continual technical training on safety is therefore one of the potential control measures to improve safety in ASGM in Zimbabwe. Pact developed a user-friendly training handbook for ASM for Zimbabwe with specific modules on underground mine support as well as safety and health [45]. Training in ASGM could also be conducted through the Ministry of Mines and Mining Development and the Mining Institutions, in addition to NGOs. However, training is a cost that requires a budget. Furthermore, the migratory nature of the unregistered ASG miners and the lack of sufficient capital for training and implementation of safe mining standards could threaten the effectiveness of training initiatives. Hence, the need for formalizing, regulating, and financing the sector is discussed further below.

Mitigation controls are relevant to Zimbabwe’s ASGM. Mine collapses and underground trappings were reported, which confirms previous literature [7,12,17]. Although the odds of experiencing accidents and injuries are generally higher in ASGM compared to LSM [6], ASGM miners in Zimbabwe are exposed to accidents and injuries with limited access to health care services and medical and social security insurance coverage [11,12,15]. Research on small enterprises involving young workers has shown that the cost of occupational injuries, diseases, and deaths on the employee (ASGM miner), the employer, and the community is 77%, 5%, and 18%, respectively [23]. Risk management reduces the time lost to injuries and associated medical costs with a positive impact on maximizing economic benefits in mining [22]; hence, improving the safety of ASGM miners is likely to be associated with improved production and positive safety outcomes. Therefore, all concerned parties will benefit from improved comprehensive risk management in ASGM in Zimbabwe. An expanded health and social security insurance coverage for Zimbabwe’s ASG miners is therefore required to reduce the adverse impacts of accidents and injuries for the miners, and the mining communities [12].

The study revealed opportunities and threats for safety management in ASGM. The majority of the participants had heard of PPE use. Training on PPE, routine inspections, management, and external accident investigations were found. Furthermore, there were positive attempts toward safe practices by a few miners. Hence, the opportunity for safety and health management. Current efforts to improve safety in ASGM focus on strengthening PPE. However, PPE is the least effective control measure for mine safety [26], as PPE cannot mitigate the risk of a mine collapse [12,18]. In addition to PPE use, shaft support is a priority for underground excavations [18]. Although the mining regulations for Zimbabwe stipulate safety standards [36], which can result in successive layers [27] of the hierarchy of controls [26], compliance was limited in ASGM [11,12,15] because of the low financial capacity and the dynamic nature of the sector [37,38]. Formalisation and regulation [39] of the sector are therefore prerequisites for the effective implementation of relevant regulations. In Mongolia for example, the ASM legal framework paved the way for the formalization of ASM, the enforcement of relevant health and safety regulations, as well as the implementation of global initiatives such as the *Fairmined gold initiative* and responsible mining standards that strengthened risk management and allowed the introduction of successive control measures, thereby providing the opportunity for comprehensive protection in ASGM [43,44]. Hence, the relevance of the ASM legal framework that addresses the safety needs of the ASGM sector in Zimbabwe.

### Limitations and Strengths

Data from the questionnaires were self-reported; recall bias, response bias, and social desirability were inevitable. Participants with low levels of literacy who were not comfortable filling in the questionnaires with support were not included in the study. Odds ratios on workplace roles did not account for confounding factors such as stress and individual health status. Data should therefore be interpreted with caution. Concurrently, self-reporting has been used successfully in social research [46] to inform decision-making and guide further research. This study, therefore, gives an overview of the prevalence of accidents and injuries and the associated risk factors. This overview could guide relevant initiatives and further research in Zimbabwe’s ASGM, and ASGM in general. The study could therefore contribute substantially to ongoing research on safety in ASGM, both locally in Zimbabwe and globally.

## 5. Conclusions

Artisanal and small-scale gold mining (ASGM) miners in Zimbabwe are faced with occurrences of accidents and injuries. Accidents and injuries were associated with underground mining, long working hours, being 18–35 years old, and being male. Underground mining was associated with high-risk activities such as blasting and the transportation of workers and materials to and from underground shafts, which was common with more mechanized mining methods. Alcohol consumption at work and drug abuse was reported. Participants had heard of PPE. There was training and inspection on PPE use. Few individual miners were introducing shaft support in hazardous ways and/or without technical support. Safety practices including shaft support, accident investigation, hazard identification, and risk assessment were missing. Mining regulations, which would provide for a range of control and safety measures, could not be adapted to the ASGM method in Zimbabwe.

### Recommendations

As indicated by Smith, ASGM should be prioritized as a high-risk sector [30] and we recommend: (i) Relevant and adaptable health and safety regulations for ASGM in Zimbabwe. (ii) Formalisation, regulation, and relevant financing schemes to improve safety in ASGM. (iii) Interventions on raising awareness on risk factors and benefits of safety practices. (iii) The identified opportunities can facilitate training on effective control measures through establishing demonstration sites. (iii) The Ministry of Health and Child Care should prioritize out-scaling national public health interventions on counseling and raising awareness of the adverse impacts of alcohol consumption and drug use in ASGM communities. (iv) ASGM associations, ASGM miners, and relevant stakeholders need to advocate for health insurance and social security for ASGM.

## Figures and Tables

**Figure 1 ijerph-19-08663-f001:**
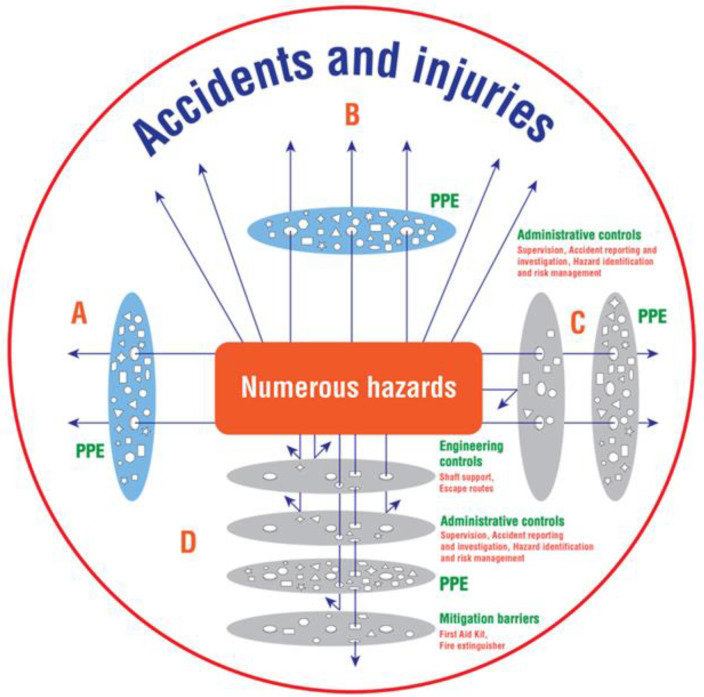
Numerous hazards and successive defensive layers: Risk management and accident prevention in ASGM in Zimbabwe modified after [12]. Key: High exposure to hazards: A and B; Single layer of control measures, e.g., PPE; Reduced exposure to hazards: C; Double layer of control measures, e.g., PPE, administrative controls; Comprehensive protection: D; Successive layers of control measures, e.g., engineering controls, administrative controls, PPE, and mitigation barriers.

**Figure 2 ijerph-19-08663-f002:**
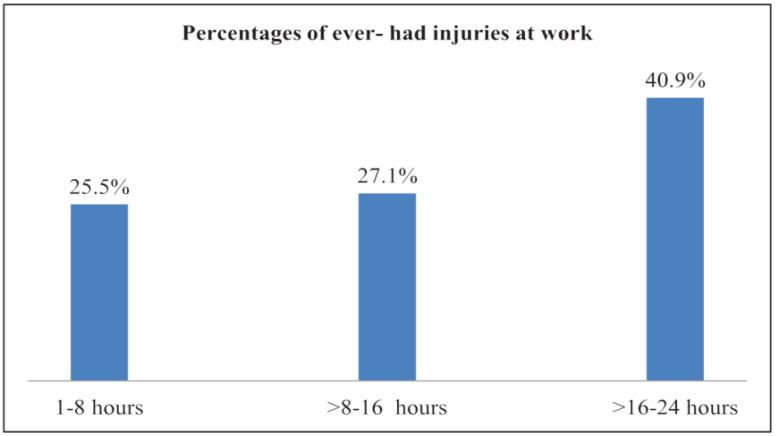
Injuries and daily working hours: Percentages of ever-had injuries at work according to daily working hours reported by miners in Kadoma and Shurugwi, Zimbabwe, in 2020. Total responses: 1–8 h = 243, >8–16 h = 70, and >16–24 h = 44.

**Figure 3 ijerph-19-08663-f003:**
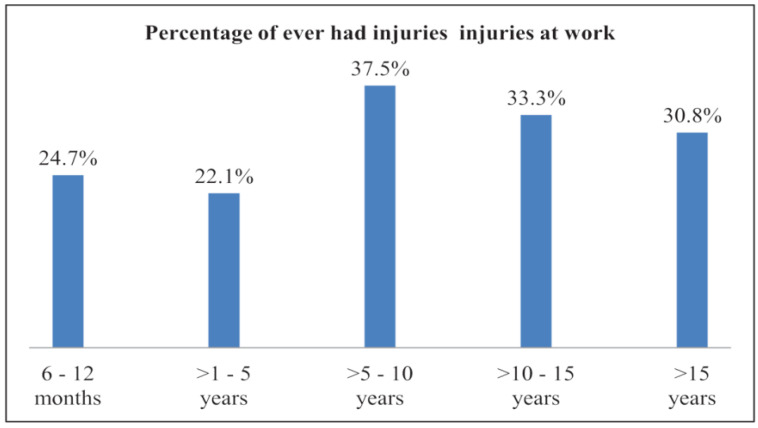
Injuries and years of experience: Percentage of injuries according to experience in ASGM reported by miners in Kadoma and Shurugwi, Zimbabwe, in 2020. Total responses 6–12 months = 93, >1–5 years = 140, >5–10 years = 48, >10–15 = 42, and >15 years = 13.

**Figure 4 ijerph-19-08663-f004:**
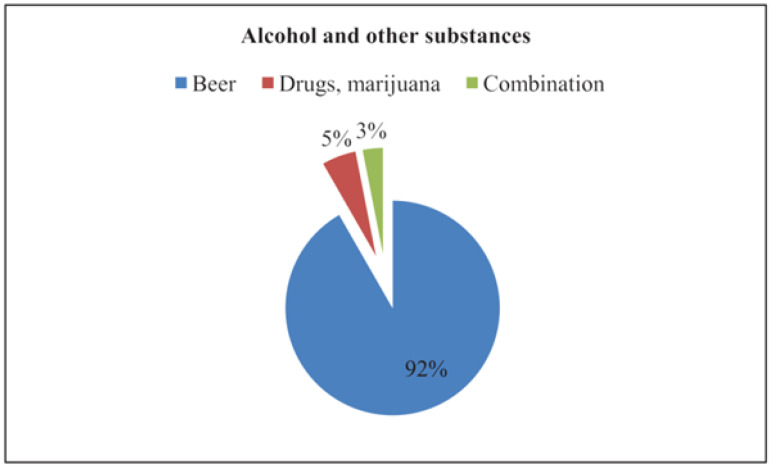
Alcohol and drug consumption: distribution of alcohol and drug use reported by ASG miners in Kadoma and Shurugwi in the 2020 rainy season, *n* = 97.

**Table 1 ijerph-19-08663-t001:** Socio-demographics: Socio-demographic characteristics of the miners from Kadoma and Shurugwi in Zimbabwe in 2020 (*n* = 401), modified [12].

Characteristics		*n* (%)	Total
Population per district		401 (100)	401
District *n* (%)	Kadoma	220 (54.9)	
	Shurugwi	181(45.1)	401
Mine category *n* (%)	Rudimentary	78 (19.5)	
	More mechanized	323 (80.5)	401
Sex *n* (%)	Female	69 (17.2)	
	Male	332 (82.8)	401
Marital status *n* (%)	Single	126 (31.9)	
	Married	202 (51.1)	
	Separated	17 (4.3)	
	Divorced	28 (7.1)	
	Widowed	22 (5.6)	395
Age *n* (%)	18–35 years	212 (56.1)	
	36–50 years	130 (34.4)	
	Above 50 years	36 (9.5)	378
Education level *n* (%)	No formal school	28 (7.1)	
	Primary	59 (14.9)	
	Secondary	241 (60.9)	
	Tertiary	39 (9.8)	
	Vocational	29 (7.3)	396
Monthly Earnings *n* (%)	No Earnings	7 (1.9)	
	Less than $100 USD	212 (56.7)	
	Above 100–500 USD	13 (34.8)	
	Above 500–1000 USD	24 (6.4)	
	Above 1000 USD	1 (0.3)	374
Roles *n* (% of cases)	Digging	211 (65.3)	
	Moving ore manually	59 (18.3)	
	Blasting	51 (15.8)	
	Loading	44 (13.6)	
	Washing/panning	33 (10)	
	Cooking	26 (7.9)	
	Amalgam burning	24 (7.3)	
	Milling	24 (7.3)	
	Sponsoring	22 (6.8)	
	Supervision	22 (6.8)	
	Mine owner	19 (5.9)	
	Gold buying	14 (4.3)	549 (Total cases)
Daily working hours *n* (%)	1–8 h	259 (66.9)	
	Above 8–16 h	82 (21.2)	
	Above 16–24 h	46 (11.9)	387
Working underground *n* (%)	Working underground yes	201(52.3)	385
Experience in ASGM *n* (%)	6–12 months	98 (26.7)	367
	>1–5 years	152 (41.4)	
	>5–10 years	57 (15.5)	
	>10–15 years	45 (12.3)	
	>15 years	15(4.1)	
Migration		112 (27.9)	394

**Table 2 ijerph-19-08663-t002:** Reported types of accidents and associated occurrences of injuries.

Type of Accident	Number (N)	Percentage of Cases	Percentage of Injuries (N)
Slips, trips, and falls (STFs)	43	40.2	52.6(20)
Hit by tools or machines	23	21.4	40.9 (9)
Hit by pieces of stone	28	26.2	50.0(13)
Breaking of winch rope	12	11.2	54.5 (6)
Collapsing	12	11.2	20.0 (2)
Mineshaft collapses	16	15.0	53.3 (8)
Underground trappings	06	5.6%	80.0 (4)

**Table 3 ijerph-19-08663-t003:** Ever-experienced injuries: Percentages of participants who ever experienced injuries and their workplace roles—as reported by artisanal small-scale gold (ASG) miners from Kadoma and Shurugwi in Zimbabwe in the 2020 rainy season, *n* = 370.

Role	Total	Ever been Injured		Crude OR (95% CI)	*p*-Value
		**Number**	**(%) ^†^**		
	370	103	25.7%		
Digging					
Yes	196	68	34.7	2.1 (1.3–3.4)	0.02 **
No	174	35	20.1	Reference	
Blasting					
Yes	45	18	40.0	1.8 (0.9–3.6)	0.05 **
No	325	85	26.2	Reference	
Washing/Processing					
Yes	30	10	33.3	1.3 (0.6–3.0)	0.5 (ns)
No	339	93	24.7	Reference	
Moving ore Manually					
Yes	55	27	49.1	3.0 (1.7–5.5)	<0.0001 **
No	315	76	24.1	Reference	
Loading					
Yes	38	18	47.4	2.6 (1.3–5.2)	0.007 **
No	332	85	25.6	Reference	
Sponsoring					
Yes	18	6	33.3	1.3 (0.5–3.6)	0.6 (ns)
No	352	97	27.6	Reference	
Manager/Supervisor/Gang leader					
Yes	20	5	25.0	0.9 (0.3–2.4)	0.7 (ns)
No	350	98	28.0	Reference	
Working at the Mill					
Yes	22	8	36.4	1.5 (0.6–3.7)	0.4 (ns)
No	348	95	27.3	Reference	
Mine Owner					
Yes	15	4	26.7	0.9 (0.3–3)	0.9 (ns)
No	355	99	27.9	Reference	
Amalgam Burning					
Yes	20	7	35.0	1.4 (0.6–3.7)	0.5 (ns)
No	350	96	27.4	Reference	
Cooking					
Yes	22	5	22.7	0.8 (0.3–2.1)	0.6 (ns)
No	348	98	28.8	Reference	
Gold Buying					
Yes	13	5	27.5	1.7 (0.5–5.2)	0.4 (ns)
No	357	98	30.2	Reference	

OR = crude odds ratio; CI = two-sided confidence interval; ^†^ = row percentages; ** = statistically significant. ns = non-significant, two-sided chi-square test.

**Table 4 ijerph-19-08663-t004:** Ever-been injured at work and exposure to risk factors: Association between ever being injured at work and exposure to risk factors, reported by miners in Kadoma and Shurugwi in Zimbabwe in the 2020 rainy season.

Characteristic	Total	Ever been Injured at Work		OR (95% CI)	AOR = (95% CI)	*p*-Value
		**Number**	**(%) ^†^**			
	370	103	(25.7)			
Sex (*n* = 370)						
Male	311	96	(30.9)	1.8 (1.02–3.3) **	4.3 (1.4–13.6)	0.01 **
Female	59	7	(11.5)	Reference	Reference	
Age (*n* = 349)						
>50	31	5	(16.1)	Reference	Reference	
36–50	125	33	(26.4)	0.8(0.5–1.3)	0.7(0.4–1.2)	0.2 (ns)
18–35	193	61	(31.6)	0.4(0.2–1.1)	0.2 (0.07–0.9)	0.03 **
Shaft miners’ transportation (*n* = 356)						
Yes	38	22	(57.9)	4.5(2.3–9) **	4.9(2.1–11.2)	<0.001 **
No	318	74	(23.3)	Reference	Reference	
Crushing (*n* = 356)						
Yes	21	14	(66.7)	6.1(2.4–16) **	9.4(2.6–34.0)	0.001 **
No	335	82	(24.5)	Reference	Reference	
Blasting (*n* = 356)						
Yes	17	10	(58.8)	4.2(1.6–11.3) **	9.2(2.6–33.0)	0.001 **
No	339	86	(25.4)	Reference	Reference	
Flying stone particles (*n* = 350)						
Yes	26	13	(26.9)	2.7(1.2–6.1) **	2.1(0.5–8.1)	0.3 (ns)
No	324	87	(31.7)	Reference	Reference	
Removing ore from the shaft (*n* = 356)						
Yes	14	4	(28.6)	1.1(0.3–3.6)	0.04(0.005–0.3)	0.002 **
No	342	92	(26.9)	Reference	Reference	
Working tools and machines (*n* = 350)						
Yes	22	9	(40.9)	1.8(0.7–4.4)	2.2(0.6–8.4)	0.3 (ns)
No	328	91	(27.3)	Reference	Reference	

AOR = adjusted odds ratio; CI = two-sided confidence interval; ^†^ = row percentages; ** = statistically significant. ns = non-significant, two-sided chi-square test.

## Data Availability

Data are available at: Singo, Josephine (2022), “Health Challenges and Risk Factors in ASGM in Zimbabwe: 2020 Survey”, Mendeley Data, V1, doi:10.17632/55vx7wjwhn.1. [47].

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
