# Peer review of "Accidents, Injuries, and Safety among Artisanal and Small-Scale Gold Miners in Zimbabwe"

_ijerph, 2022, doi:10.3390/ijerph19148663_

Round 1

Reviewer 1 Report

Underground mining is a risky activity all around the world, but the risks are higher in under-developed countries. So, focus on the prevention of accidents in this field is welcome.

Observations:

1. Figure 1 is too large. It should be reduced at 50%.

2. The questionnaire is an important part of the study and should be attached to this article. If you consider that it is too long, you should consider presenting only its basic structure and, in this case, it could be placed in the body of the article, not as an appendix.

When I tried to get it using the DOI of reference [12], I received the message that the DOI is wrong. Please correct it.

3. At page 4, there is a manual hyphenation “purpos-ively”. Please correct it. Also, “par-ticipants” on the same page.

4. At page 5, there is actually Figure 2, not Figure 1. This figure should be properly placed in page. Now it reaches the right margin of the paper.

5. At page 8, the figure is too large. Please reduce it at 50%. (Same for figures at pages 9 and 12.)

6. At page 8, there is a paragraph: “Of the miners who had experienced workplace violence, 39.4% (n=37) had ever experienced injuries at work (OR= 2.1 [1.3–3.4] p=005), which complements findings from

the same survey [12].” It is not clear what was the relevance of the workplace violence. And because it is mentioned at page 13 that “Workplace violence is not discussed in this article.”, maybe the paragraph at page 8 should be removed.

7. At page 12, there is not presented a clear-direct connection between alcohol and drug use and accidents (or types of accidents). (That there is a regrettable culture of alcohol and drug consumption, there is no doubt.)

8. At page 15, the recommendations are too brief, too generic. They should be expanded. For example, what is the recommended approach to prevent alcohol and drug consumption at workplace? - briefing the workers about the risks or imposing sanctions on the team leader for accepting drunk workers on site?

Author Response

Thank you very much for the useful observations.

Please find attached our responses.

Kind regards,

Josephine

Reviewer 2 Report

Review report for the paper “Accidents, injuries, and associated risk factors among artisanal and small-scale gold miners in Zimbabwe”

Why do we need this study? I did not see the author discussing the reason. Therefore, it is impossible to prove it. Need detailed further explanation.

Insufficient expression on innovative explanations. Does the practical significance of this innovation exist? There is a lack of comparison with previous studies of the same kind. For this point, the innovativeness of the author's statement needs further explanation.

Indicator issues. Is it appropriate for the author to directly use research results in similar literatures into the research questions of this article? Is there a better reference standard in similar studies? In the subdivision question of this article, do you need to further improve the research results of other scholars in the index design? Please give a reliable argument for the indicator design.

Literature review. Add more recent papers published in last three years. Remove papers published before 2017. Based on the LR you should define the scientific gap. I suggest authros to read and discuss following papers: Simić, N., Stefanović, M., Petrović, G., & Stanković, A. (2021). Use of the risk analysis approach n the Serbian army integration process against Covid-19. Operational Research in Engineering Sciences: Theory and Applications, 4(1), 67-81. https://doi.org/10.31181/oresta2040127s; Supriyatna, H., Kurniawan, W., & Purba, H. H. (2020). Occupational safety and health risk in building construction project: Literature review. Operational Research in Engineering Sciences: Theory and Applications, 3(1), 28-40. https://doi.org/10.31181/oresta200134s; Kabir, M. F., & Roy, S. (2021). Hazard perception test among young inexperienced drivers and risk analysis while driving through a T-junction. Decision Making: Applications in Management and Engineering. https://doi.org/10.31181/dmame181221015k; Komazec, N., Mladenović, M., & Dabižljević, S. (2018). Etiology of the notion of event in terms of decision-making and determination of organizational system risk conditions. Decision Making: Applications in Management and Engineering, 1(1), 165-184. https://doi.org/10.31181/dmame1801165k.

There is no comparative proof, no analysis of the superiority of the proposed methodology with other approaches in the literature. Lack of comparison of results under different models.

In the part of research status, the outline of the whole research is not clear enough, and more content of multi criteria decision model (method) needs to be added.

The results of the application part of the model need to be rearranged, the readability is too poor, and the graphical results provided can’t make people see the differences under different scene settings.

Author Response

Why do we need this study? I did not see the author discussing the reason. Therefore, it is impossible to prove it. Need detailed further explanation.

We have given the reason based on the context of high prevalence of accidents and injuries and little research on the topic

Insufficient expression on innovative explanations. Does the practical significance of this innovation exist? There is a lack of comparison with previous studies of the same kind. For this point, the innovativeness of the author's statement needs further explanation.

We gave further explanation with reference to literature. Comparison has been given with reference to Kenya, DRC, Ethiopia, Mongolia etc. 

Indicator issues. Is it appropriate for the author to directly use research results in similar literatures into the research questions of this article? Is there a better reference standard in similar studies? In the subdivision question of this article, do you need to further improve the research results of other scholars in the index design? Please give a reliable argument for the indicator design.

Reference to previous studies has been given as mentioned above. 

Literature review. Add more recent papers published in last three years. Remove papers published before 2017. Based on the LR you should define the scientific gap. I suggest authros to read and discuss following papers: Simić, N., Stefanović, M., Petrović, G., & Stanković, A. (2021). Use of the risk analysis approach n the Serbian army integration process against Covid-19. Operational

Research in Engineering Sciences: Theory and Applications, 4(1), 67-81. https://doi.org/10.31181/oresta2040127s; Supriyatna, H., Kurniawan, W., & Purba, H. H. (2020). Occupational safety and health risk in building construction project: Literature review.

Operational Research in Engineering Sciences: Theory and

Applications, 3(1), 28-40. https://doi.org/10.31181/oresta200134s; Kabir, M. F., & Roy, S. (2021). Hazard perception test among young inexperienced drivers and risk analysis while driving through a T-junction. Decision Making: Applications in Management and

Engineering. https://doi.org/10.31181/dmame181221015k; Komazec, N., Mladenović, M., & Dabižljević, S. (2018). Etiology of the notion of event in terms of decision-making and determination of organizational system risk conditions. Decision Making:

Applications in Management and Engineering, 1(1), 165-184.

https://doi.org/10.31181/dmame1801165k.

 We added more  references relevant to develop a more comprehensive theoretical frame work and we have edited our conceptual framework accordingly. However, research on accidents and injuries in ASGM is limited. Deleting all references before 2017 was therefore not possible. In addition our conceptual framework is based on literature published before 2017. The majority of the background research on ASGM in Zimbabwe was published before 2017. We have also checked the most recent publications on ASGM in IJERPH and we have noted that the references included literature which was published before 2017: https://researchonline.lshtm.ac.uk/id/eprint/4663761/1/ijerph-18-11031.pdf

There is no comparative proof, no analysis of the superiority of the proposed methodology with other approaches in the literature.

Lack of comparison of results under different models.

The focus of the article was on prevalence of accidents and injuries and protective measures, and we applied the concept of the Swiss Cheese Model and the hierarchy of controls.

In the part of research status, the outline of the whole research is not clear enough, and more content of multi criteria decision model (method) needs to be added.

We improved clarity focusing on the Swiss cheese model and the hierarchy of control measures. The article was focusing on the existing protective barriers Engineering controls, administrative controls, PPE, mitigation controls.  Our data did not include decision making, decision models were therefore not included.

The results of the application part of the model need to be rearranged, the readability is too poor, and the graphical results provided can’t make people see the differences under different

scene settings.

Results have been presented under subtitle. We have presented three contexts in the model: high exposure to hazards, limited exposure and comprehensive protection

Thank you very much for the useful observations.

Please find attached our responses.

Kind regards,

Josephine

Reviewer 3 Report

Thank you very much for providing me an opportunity to read and review the paper entitled “Accidents, injuries, and associated risk factors among artisanal and small-scale gold miners in Zimbabwe”. By using data from 401 participants, this research papers tries to know how risk factors may impact on the prevalence of accidents and injuries. Interestingly in the context of Zimbabwe a country in southern Africa  they found that injuries were associated with underground mining risks. Overall, it’s an interesting research topic. The context of study makes the study finding more interesting. I hope my comments help you to improve the paper and make it publishable. 

My first recommendation to the authors of paper is to remove all numbers from the abstract of the paper. For instance, instead of “workplace roles associated with experiencing injuries were digging (Odds Ratio [OR] = 2.1 [1.3–3.4]), blasting (OR= 1.8 [0.9–3.6]), moving ore manually (OR= 3.0 [1.7–5.5]), and loading (OR= 2.6 [1.3–5.2])”, you may use “workplace roles associated positively with experiencing injuries were digging, blasting, moving ore manually, and loading”. 

I would like to thank you the authors of the paper, for keeping the method and data collection transparent. They clearly mentioned that they have used a part of their data for another published paper entitled: “Hazards and control measures among artisanal and small-scale gold miners in Zimbabwe”.  My academic background is business and management, I have done some research in context of mental health of workers during Covid pandemic as well. Its not very common in our field this type of transparency. 

As far as I know Zimbabwe it has an adult literacy rate of 88.69%. This rate among mining workers should be even lower. During your data collection, how you treat peoples who were unable to write and read? Since you used random sampling method for selecting your final sample. Most probably, some part your study’s population were illiterate or were unable to understand clearly your research question. For increasing the transparency of your research, I recommend you to explain with detailed information your selection process of final sample.  

A minor comment: the size of your figures is large, it’s much better if you reduce little bit their size

Author Response

(The authors gave the same response as above.)

Round 2

Reviewer 2 Report

The authors have addressed the point of my concern. I am happy with their corrections. Hence, I would like to recommend this manuscript to be published.